



# Statistical analysis of Lagrangian transport of subtropical waters in the Japan Sea based on AVISO altimetry data

Sergey V. Prants, Maxim V. Budyansky, and Michael Yu. Uleysky

Laboratory of Nonlinear Dynamical Systems, Pacific Oceanological Institute of the Russian Academy of Sciences, 43 Baltiyskaya st., 690041 Vladivostok, Russia, URL: http://dynalab.poi.dvo.ru

*Correspondence to:* S.V. Prants
(prants@poi.dvo.ru)

**Abstract.** Northward near-surface Lagrangian transport of subtropical waters in the Japan Sea frontal zone is simulated and analyzed based on altimeter data for the period from January 2, 1993 to June 15, 2015. Computing different Lagrangian indicators for a large number of synthetic tracers launched weekly for 21 years in the southern part of the Sea, we find preferred transport pathways across the Subpolar Front. This cross-frontal transport is statistically shown to be meridionally inhomo-
geneous with "gates" and "barriers" whose locations are determined by the local advection velocity field. The gates "open" due to suitable dispositions of mesoscale eddies facilitating propagation of subtropical waters to the north. It is documented for the western, central and eastern gates with the help of different kinds of Lagrangian maps and verified by some tracks of available drifters. The transport through the gates occurs by a portion-like manner, i.e., subtropical tracers pass the gates in specific places and during specific time intervals. There are some "forbidden" zones in the frontal area where the northward
transport has not been observed during all the observation period. They exist due to long-term peculiarities of the advection velocity field there.

## 1 Introduction

The Japan Sea (JS) is a mid-latitude marginal sea with dimensions of $1600 \times 900$ km, the maximal depth of 3.72 km and the mean depth of about 1.5 km. It spans regimes from subarctic to subtropical and is characterized by many of the same
phenomena found in the deep ocean: fronts, eddies, currents and streamers, deep water formation, convection and subduction. It communicates with the Pacific Ocean at the south and east through the Tsushima/Korean and Tsugaru straits, respectively. In the north it is connected with the Okhotsk Sea through the Soya (La Perouse) and Tatarsky straits. All the four channels are shallow with depths not exceeding 135 m.

Bathymetry of the JS and its geographic and oceanographic features are shown in Fig. 1S in Supplementary material. Warm
and saline Pacific waters enter the Tsushima Strait and splits into three currents (Fig. 1). The Nearshore Branch of the Tsushima Current flows northward along the western coast of the Honshu Island (Japan). Its Offshore Branch with a meander-like path flows into the Yamato Basin. The East Korean Warm Current flows northward along the eastern coast of Korea to meet the North Korean Cold Current which is a prolongation of the Liman Cold Current flowing southward along the Siberian coast down to Vladivostok. One of the major large-scale feature in the northern JS is a cyclonic gyre over the Japan Basin and the



Tatarsky Strait. Some aspects of the surface circulation in the JS have been studied by Hirose et al. (2005); Lee and Niiler (2005); Danchenkov et al. (2006); Talley et al. (2006); Yoon and Kim (2009); Kim and Yoon (2010); Lee and Niiler (2010); Ito et al. (2014).

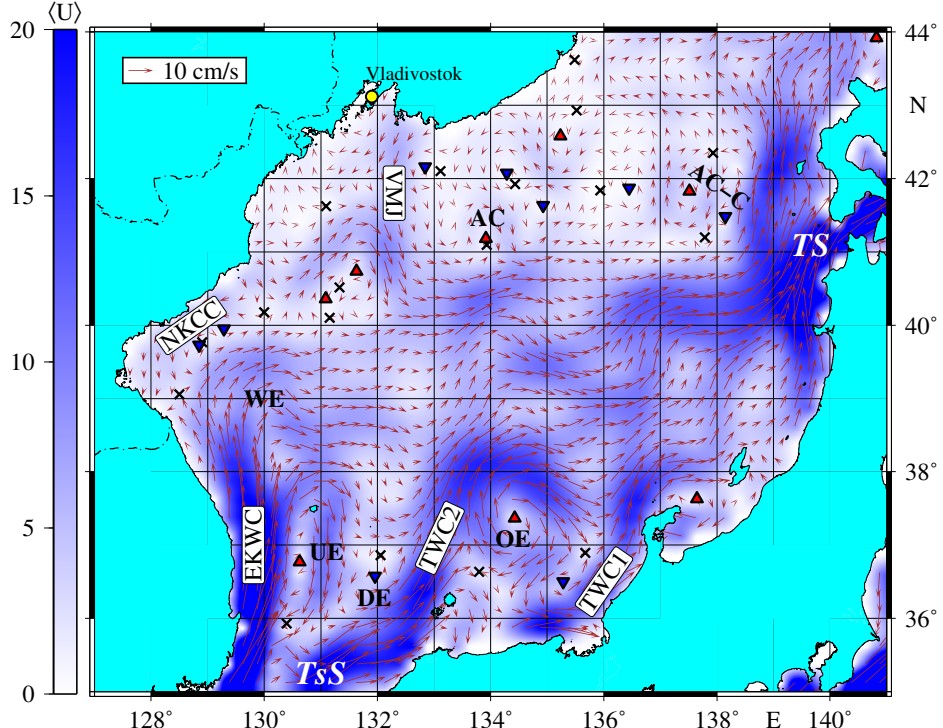

**Figure 1.** The AVISO velocity field averaged for the period from January 2, 1993 to June 15, 2015. Elliptic and hyperbolic stagnation points with zero mean velocity are indicated by triangles and crosses, respectively. Abbreviations: TsS (Tsushima or Korean Strait), TS (Tsugaru Strait), EKWC (East Korean Warm Current), NKCC (North Korean Cold Current), TWC1 and TWC2 (the first and second branches of the Tsushima Warm Current, UE (Ulleung eddy), DE (Dok eddy), OE (Oki eddy), WE (Wonsan eddy), AC-C (vortex pair near the eastern gate), AC (anticyclonic eddy over the Japan Basin), VMJ (Vladivostok meridional jet).

The confluence of northward warm subtropical waters with southward cold subarctic ones forms one of the most remarkable features in the Sea, the distinct SF that extends across the basin near $40°$ N (Park et al., 2004; Talley et al., 2006). It is a boundary of physical and chemical properties such as temperature, salinity, dissolved oxygen and nutrients. Like many other hydrological fronts, the SF is a highly productive zone with favorable fishery conditions. It is not a continuous curve crossing the basin with a maximal thermal gradient. It is rather a vast area between $38°$ N and $41°$ N extending across the basin from the Korea coast to the Japanese islands.

Understanding transport pathways of subtropical water in the JS is relevant to a number of applications. Physical properties (temperature and salinity), chemical properties, pollutants and biota (phytoplankton, zooplankton, larvae, etc.) are transported



and mixed by currents and eddies. Transport of heat to the north is crucial for climatic applications. The ability to simulate transport adequately would be useful to deal with the aftermath of accidents at sea such as discharges of radionuclides, pollutants and oil spills. It is also crucial, for instance, for understanding transport pathways for species invasions.

Since the last decades in the twentieth century, invasions of heat-loving fish (conger eel, tuna, moonfish and triggerfish) and some tropical and subtropical marine organisms (turtles, sharks and others) have been observed in the northern part of the Sea, to the coast of Russia (Ivankova and Samuilov, 1979). It is natural to assume that such invasions could be caused by intrusions of subtropical waters in the northern part of the Sea across the SF. They may be also one of the reasons for a prolongation of the warm period in the fall in Primorye province in Russia since the 1990s (Nikitin et al., 2002). From the oceanographic point of view, this transport of subtropical waters contradicts long-held beliefs on circulation in the JS. It is believed that the SF is a transport barrier for propagation of subtropical waters across it to the north, at least, in the western and central parts of the front area (see, e.g., Danchenkov et al., 2006). In this paper we use the altimetry data to simulate and analyze the northward near-surface transport of subtropical waters across the frontal area from January 2, 1993 to June 15, 2015.

The paper is organized as follows. Section 2 introduces briefly the altimetry data and simulation methods we use. Northward transport of subtropical waters across the SF area is studied statistically in Sec. 3 based on the altimetry data for a long period of time. We compute, document and discuss preferred transport pathways and meridional distributions of artificial tracers launched in the southern part of the Sea. A summary of the main results obtained is presented in Sec. 4. Supplementary data, associated with this article, can be found in the on-line version.

## 2   Data and methods

Geostrophic velocities were obtained from the AVISO database (http://aviso.altimetry.fr) archived daily on a $1/4° × 1/4°$ grid from January 2, 1993 to June 15, 2015. Our Lagrangian approach is based on solving equations of motion for a large number of passive synthetic particles (tracers) advected by the AVISO velocity field

$$\frac{d\lambda}{dt} = u(\lambda, \varphi, t), \qquad \frac{d\varphi}{dt} = v(\lambda, \varphi, t), \tag{1}$$

where $u$ and $v$ are angular zonal and meridional velocities, $\varphi$ and $\lambda$ are latitude and longitude, respectively. Bicubical spatial interpolation and third order Lagrangian polynomials in time are used to provide numerical results. Lagrangian trajectories are computed by integrating the equations (1) with a fourth-order Runge-Kutta scheme with an integration step to be $1/1000$ day. Lagrangian analysis of transport and mixing in marginal seas and in the deep ocean has experienced intense developments in the last decade (Harrison and Glatzmaier, 2012; Huhn et al., 2012; Prants et al., 2011b; Hernández-Carrasco et al., 2011; Keating et al., 2011; Prants, 2013, 2014; Budyansky et al., 2015; Rossi et al., 2013; Prants, 2015).

The merged TOPEX/POSEIDON and ERS-1/2 altimeter data sets have been shown by Choi et al. (2004) to be appropriate to study mesoscale surface ocean circulation in the JS because of their comparatively small temporal and spatial sampling intervals. In particular, they have been shown to correlate well (0.95) with tide gauge data in the western JS (Choi et al., 2004). We would like to stress that the AVISO velocity field, averaged for the period from January 2, 1993 to June 15, 2015 and shown





in Fig. 1, demonstrates all the known mesoscale features of near surface circulation in the Sea including even correct locations of Ulleung, Dok, Oki and Wonsan quasi-permanent mesoscale eddies.

However, altimetry data provide the velocity field which is a geostrophical approximation to the real near-surface velocities. The results of our altimetry-based Lagrangian statistical analysis are expected to be valid on a mesoscale where the AVISO field may be considered to be a good approximation. Altimetry-based results should be considered with a caution when dealing with submesoscale structures (Keating et al., 2011). Local submesoscale phenomena like frontogenesis and ageostrophic instabilities cannot be reproduced correctly in altimetry-based velocity fields.

Transport of tracers is simulated for a comparatively long period of time, up to two years, and our results are based not on individual trajectories but on statistics for hundreds of thousands of trajectories. We cannot, of course, guarantee that we compute "true" trajectories for individual tracers in a chaotic velocity field. However, the description of general pattern of transport for thousands of tracers is much more robust. The shadowing lemma (see, e.g., Ott, 2002) states that although a numerically computed chaotic trajectory diverges exponentially from the "true" trajectory with the same initial conditions, there exists "a true" trajectory with a slightly different initial condition that stays near the numerically computed one. In other words, nobody is able to reproduce motion of a single passive particle in a chaotic flow, but it is possible to reproduce transport of statistically significant number of particles.

We study northward transport of tracers in the central part of the JS basin between $37°$ N and $42°$ N. With this aim $10^5$ tracers have been launched weekly from January 2, 1993 to June 15, 2013 at the latitude $37°$ N from $129°$ E to $138°$ E. Trajectory of each tracer has been computed for two years after its launch date. We fixed the location and the moment of time where and when each tracer crossed a given latitude in the central JS between $37°$ N and $43°$ N. We fix only the first crossing of a given zonal line. We take into account the first passage only, because we are interested not in a net transport but in the northward transport only. We stop to compute trajectories of those tracers which get into an AVISO cell with at least two corners situated at the land.

To simulate and analyze transport across the frontal area, we propose a Lagrangian methodology with a complex of the numerical codes compiled to compute a number of Lagrangian indicators for synthetic tracers and plot the following diagrams and Lagrangian maps.

1) Meridional distribution of the number of tracers, $N$, crossing fixed latitudes, $\lambda_f$, in the central JS with a space step $0.1°$. The corresponding data are represented as a density map which shows by color the density of tracks of the tracers crossed all the latitudes in the central JS from January 2, 1993 to June 15, 2015. Tracking maps show where the subtropical tracers, which crossed eventually the fixed zonal line through fixed meridional "gates", wandered for the whole integration period. They also can be represented as a $N(\lambda_f)$ distribution which shows how many tracers reached a fixed zonal line at the longitude $\lambda_f$ for the whole period of integration.

2) Fixing initial longitudes $\lambda_0$ of launched tracers along the material line $37°$ N, we compute those final longitudes $\lambda_f$ at which they cross a fixed zonal line for the whole period of integration. The results are represented as $\lambda_0 - \lambda_f$ plots.

3) The $T - \lambda_f$ plots show when and at which longitudes the tracers, launched at $37°$ N, crossed the latitudes $40°$ N and $42°$ N for the whole period of integration.



4) In order to document and visualize intrusions of subtropical waters into subarctic ones we compute so-called Lagrangian maps by integrating the advection equations (1) backward in time (Prants, 2015). A subbasin in the Sea is seeded at a fixed date with a large number of tracers whose trajectories are computed backward in time for a given period of time. We use three kinds of the Lagrangian maps in this paper. Such maps have been shown to be useful in studying large-scale transport and mixing

in various basins, from bays (Prants et al., 2013) and seas (Prants et al., 2011a, 2013) to the ocean scale (Prants et al., 2011b; Prants, 2013), in quantifying propagation of radionuclides in the Northern Pacific after the accident at the Fukushima Nuclear Power Plant (Prants et al., 2011b, 2014a; Prants, 2014; Budyansky et al., 2015) and in finding potential fishing grounds (Prants et al., 2014b, c).

In order to track those subtropical waters which were able to cross the SF and reach given latitudes in the northern JS,

we mark by color the tracers that reached the line $37°$N in the past and compute how much time it took. In order to know where this or that tracer came from for a given period of time, we compute the drift maps with boundaries. The waters, that entered a given area through its southern boundary, are shown by one color on such a map, and the waters, that came through the northern boundary, are shown by another color. With another kind of maps, drift maps, we compute the finite-time displacement of tracers, $D$, that is a distance between the final, $(\lambda_f, \varphi_f)$, and initial, $(\lambda_0, \varphi_0)$, positions of advected particles on the Earth

sphere.

"Instantaneous" stagnation elliptic and hyperbolic points are indicated on the Lagrangian maps by triangles and crosses, respectively. They are points with zero velocity which are computed daily. Up(down)ward orientation of one of the triangle's top means anticyclonic (cyclonic) rotations of water around them. The triangles are colored as red (blue) marking elliptic points for anticyclones (cyclones). The elliptic points, situated mainly in the centers of eddies, are those stagnation points

around which the motion is stable and circular. The hyperbolic points, situated mainly between and around eddies, are unstable ones with the directions along which waters converge to such a point and another directions along which they diverge. The stagnation points are moving Eulerian features and may undergo bifurcations in the course of time. In spite of nonstationarity of the velocity field some of them may exist for weeks and much more.

We have used for a comparison and verification tracks of surface drifters that are available at the site http://aoml.noaa.gov/

phod/dac.

## 3 Results and Discussion

### 3.1 Northward transport of subtropical water and advection velocity field

Figure 1 with the AVISO velocity field, averaged for the period from January 2, 1993 to June 15, 2015, reflects the known features of mesoscale near-surface circulation in the central JS. The Tsushima Warm Current splits into three parts. The first

one (TWC1 in Fig. 1) is the near shore branch flowing northward along the western coast of the Honshu Island (Japan). The second one (TWC2) is the offshore meander-like branch, and the third one is the East Korean Warm Current flowing northward as a western boundary current along the eastern coast of the Korean peninsula (EKWC). It encounters the North Korean Cold





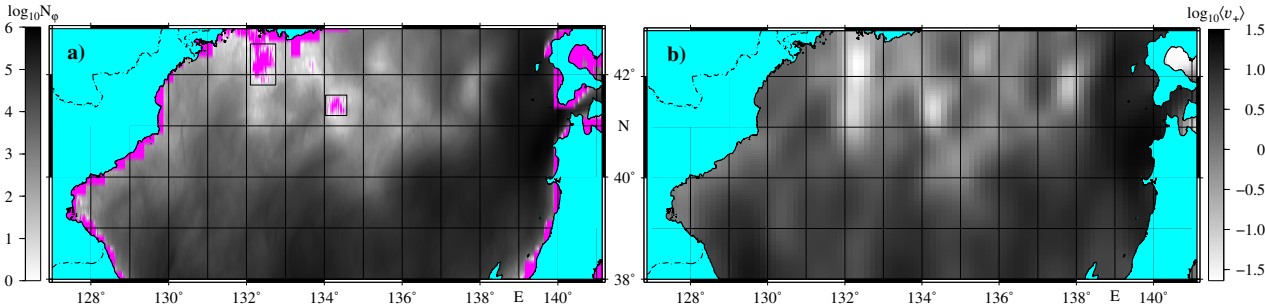

**Figure 2.** a) The logarithmic-scale density of tracks of the tracers crossing all the latitudes $\varphi$ in the central JS, $N_\varphi$, from January 2, 1993 to June 15, 2015. The rectangular magenta areas are forbidden zones where the northward transport has not been observed during the whole integration period. The other magenta areas near the cost mean that the AVISO grid cells there touch the land, and we did not compute trajectories there. The tracers have been launched weekly along the zonal line at $37°$ N from January 2, 1993 to June 15, 2013. b) Distribution of the averaged northward component of the AVISO velocity field $\langle v_+(\lambda, \varphi)\rangle$ in the logarithmic-scale averaged over the same period.

Current flowing southward (NKCC). Both the currents separate from the Korean coast at about $39°$ N and flow to the east forming the SF.

Some well known quasi-stationary mesoscale eddy-like features are also visible in Fig. 1. In the Ulleung Basin there are the warm Ulleung anticyclonic circulation (Shin et al., 2005; Mitchell et al., 2005; Shin, 2009; Lee and Niiler, 2010) with the

center at about $37°$ N, $130.5°$ E (UE) and a cyclonic circulation called often as the cold Dok eddy (DE) (Lee and Niiler, 2010) with the center at about $36.7°$ N, $132°$ E. The flow over bottom topography around the Oki Spur in the south-eastern part of the Sea generates the anticyclonic Oki Eddy (OE) with the center at about $37.5°$ N, $134.2°$ E (Isoda, 1994). In the western part of the Sea meandering of the East Korean Warm Current produces an anticyclonic circulation called as the anticyclonic Wonsan Eddy (WE) with the center at about $39°$ N, $129°$ E (Lee and Niiler, 2005).

Plot in Fig. 2a shows the density of tracks of tracers launched along $37°$ N and crossed all the latitudes in the central JS for the whole period of integration, from January 2, 1993 to June 15, 2015. The density is coded by nuances of the grey color in the logarithmic scale, $\log_{10} N_\varphi$. The magenta areas in Fig. 2a along the coastal line mean that the AVISO grid cells there touch the land, and we did not compute trajectories there. Uneven density of points in Fig. 2a means that the northward transport of subtropical waters is meridionally inhomogeneous with a kind of "gates" with increased density of points. The gates are such

spatial intervals along a given zonal line across which subtropical tracers prefer to cross it.

Any tracer, as a passive particle, is able to cross the fixed latitude in the northward direction if the northward component of the velocity field is nonzero at its location. In Fig. 2b we plot distribution of the northward component of the AVISO velocity field averaged over the whole period of integration as follows:

$$\langle v_+(\lambda, \varphi)\rangle = \frac{1}{\mathbf{n}} \sum \theta(v(\lambda, \varphi))v(\lambda, \varphi), \tag{2}$$





where $v_+(\lambda, \varphi)$ is a northward (positive) component of the velocity at the point $(\lambda, \varphi)$, $\theta(v)$ the Heaviside function and $\mathbf{n}$ the number of days in the period from January 2, 1993 to June 15, 2015. Comparing Lagrangian representation in Fig. 2a with the Eulerian one in Fig. 2b, it is clear that areas with increased density of points in Fig. 2a correlates well with areas with increased average values of the northward component of the AVISO velocity field in Fig. 2b.

Thus, the northward transport of subtropical waters in the central JS is determined mainly by the local advection velocity field, more precisely by local values of the northward component of the velocity. The greater is the northward component of the velocity at a given point and the longer is the period of time when it is positive the more tracers are able to cross the corresponding latitude.

The density difference in some meridional ranges in Fig. 2a may be very large because of the logarithmic-scale representation. There are even some places in the northern SF area where the northward transport has not been observed during all the simulation period, from 1993 to 2015. They are marked by magenta rectangles in Fig. 2a. One "forbidden" zone is situated in the deep Japan Basin with the center at about $41.5°$ N, $134.2°$ E, and another one is situated to the south off Vladivostok from $43°$ N to $41°$ N approximately along the $132°$ E meridian. We stress that they are forbidden only to northward transport of tracers but can be and really are open to transport in other directions.

The "forbidden" zones exist due to long-term peculiarities of the advection velocity field there. The zone to the south off Vladivostok exists due to a quasi-permanent southward jet approximately along the meridian $132°$ E from $43°$ N to $40°$ N (VMJ in Fig. 1). It turns to the east at about $40°$ N and contributes to the eastward transport. In fact, the northward velocity is practically zero in this area (see Fig. 2b) and, therefore, the northward transport is absent. The other "forbidden" zone exists due to two factors, the presence of a quasi-permanent anticyclonic eddy with the center at about $41.3°$ N, $134°$ E in the deep Japan Basin (AC in Fig. 1) and the eastward zonal jet blocking northward transport across it. Topographically constrained anticyclonic eddies with the center at about $41°$ N $- 41.5°$ N, $134°$ E $- 134.5°$ E have been regularly observed there (Takematsu et al., 1999; Talley et al., 2006; Prants et al., 2015).

## 3.2 Transport pathways of subtropical water and its intrusions across the Subpolar Front

Now let's look more carefully at the meridional distribution of subtropical tracers crossed the SF for the whole period of simulation. We choose for reference four zonal lines along the AVISO grid at $42.125°$ N, $129°$ E $- 141.24°$ E; $41.875°$ N, $129°$ E $- 141.18°$ E; $40.125°$ N, $128°$ E $- 140.24°$ E and at $39.875°$ N, $128°$ E $- 140.24°$ E. These distributions are shown in Fig. 3 for each zonal line by solid curves with superimposed meridional distributions of the averaged northward AVISO velocity (arrows). The numbers of crossings of those latitudes by available drifters are shown by dashed curves. The correspondence between the peaks in the meridional distributions of the tracers and of the averaged northward AVISO velocity is rather good for all the chosen zonal lines confirming their direct connection.

The tracer's peaks correlate more or less with the number of crossings of the chosen zonal lines by drifters. The correlation is rather good for the western and eastern parts but not for the central one. The comparison with drifters should be taken with care because of a comparatively small number of available drifters especially in the central part of the SF. Moreover, the drifters, are





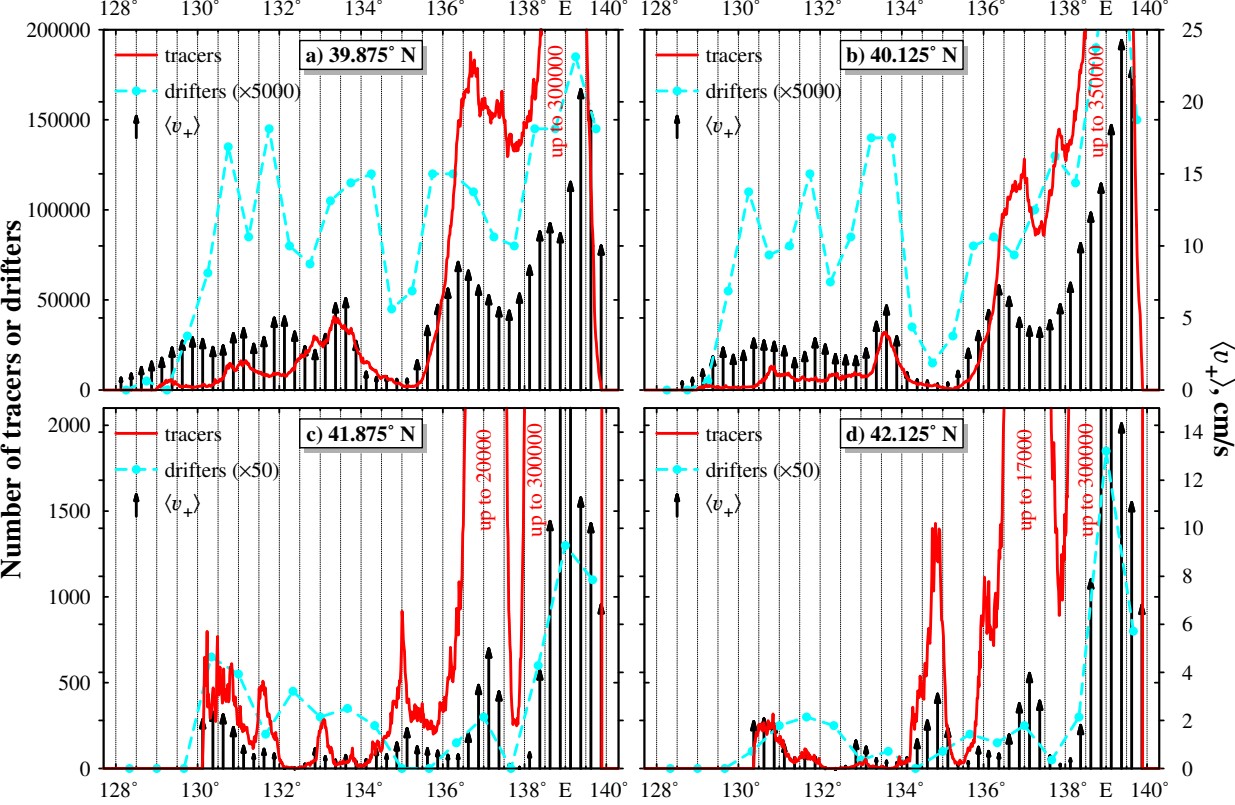

**Figure 3.** Meridional distributions of the number of tracers which crossed indicated zonal lines (solid curves), of the averaged northward component of the AVISO velocity in cm s$^{-1}$ (arrows) and of the number of crossings of those zonal lines by available drifters (dashed curves). The period of observation is from January 2, 1993 to June 15, 2015.

not ideal passive tracers and they, of course, have not been launched at the zonal line 37° N like artificial tracers in simulation. Their launch sites have been distributed rather randomly over the basin.

The meridional tracer distribution in Fig. 3 allows to distinguish the eastern, central and western gates in the central JS which strongly differ by the number of passing tracers. The very eastern, 138° E−140° E, and western, 129° E−131° E, gates are provided mainly by the near shore branch of the Tsushima Warm Current and the East Korean Warm Current, respectively. The central gate, 133° E−137° E, exists, probably, due to topographically constrained features over the Yamato Rise there (see Fig. 1S in Supplementary material). The transport through that gate will be shown to be enhanced due to a specific disposition of SF eddies regularly observed there. The intervals between the gates may be called "conditioned barriers" because of a comparatively small number of tracers crossing zonal lines there, and because they used to "open" for a comparatively short time intervals.





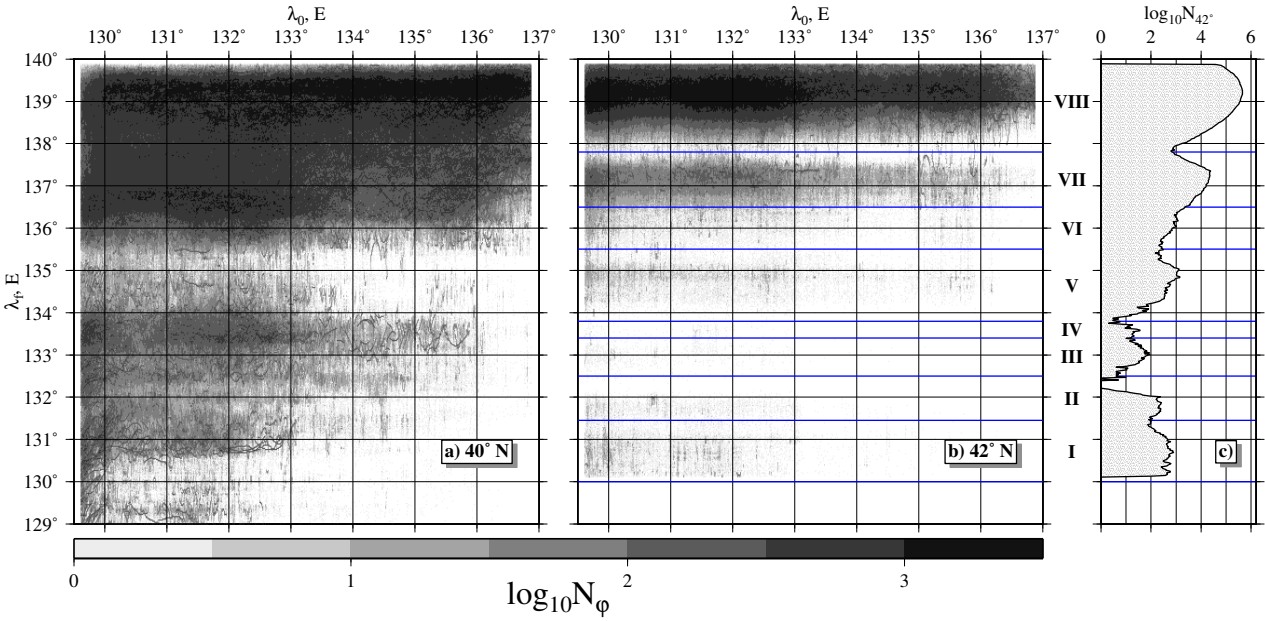

**Figure 4.** Density plots show in the logarithmic scale how many and at which final longitudes $\lambda_f$ the tracers with initial longitudes $\lambda_0$, were able to cross the zonal lines a) $40°\,$N and b) $42°\,$N for the whole simulation period. The tracers have been launched weekly at the line $37°\,$N from January 2, 1993 to June 15, 2013. c) Meridional distribution of the number of tracers which crossed the zonal line $42°\,$N for the whole simulation period. This line is divided in eight intervals numbered by the roman numerals.

Figures 4a and b show in accordance with the task 2 at which final longitudes $\lambda_f$ the tracers, launched with the initial longitudes $\lambda_0$ at the line $37°\,$N, reached the zonal lines $40°\,$N and $42°\,$N for the whole period of integration. Meridional distribution of the number of tracers with pronounced peaks which crossed the zonal line $42°\,$N for the same period is plotted in Fig. 4c. This zonal line was divided in eight meridional intervals numbered by the roman numerals in Figs. 4b and c with

5 the horizontal straight lines running via local minima at the distribution in Fig. 4c.

The Tsushima Warm Current contributes mainly to the eastern peak VIII at the distribution in Fig. 4c. The black color across all the range of initial longitudes $\lambda_0$ in Fig. 4b means that fluid particles, crossing eventually the line $42°\,$N through the gate $138°\,$E $-140°\,$E, could have any value of the initial longitude $\lambda_0$ at the zonal line $37°\,$N. They could reach that gate by different ways: either to be initially trapped by the near shore branch or to be advected by the offshore branch and then to enter the near

shore branch. Moreover, those particles could be involved initially in the East Korean Warm Current and then be transported to the east along the SF and eventually join to the Tsushima Warm Current. Thus, the subtropical tracers, crossing the gate VIII, may have rather distinct values of some Lagrangian indicators, e.g., travelling time and distance passed.

There is a narrow barrier, the white strip in Fig. 4b between the gates VIII and VII, with the center at the local minimum at $137.8°\,$E in Fig. 4c. A comparatively small number of tracers have been able to cross the line $42°\,$N there for the whole

simulation period. The gate VII between $136°\,$E and $137.8°\,$E (Figs. 4b and c) provides northward transport of subtropical

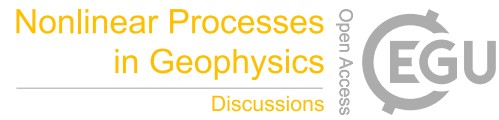



tracers by means of a quasi-permanent vortex pair located there. The number of subtropical tracers passing through this gate is much smaller than that passing through the gate VIII (remember the logarithmic scale in Fig. 4). Only a small number of tracers, launched initially at the very eastern part of the zonal line $37°$ N, were able to cross the line $42°$ N through that gate, because most of the eastern tracers passed through the gate VIII to be captured by the near shore branch of the Tsushima Warm

Current. Most of the tracers, passing through the gate VII, came from the western and central parts of the material line at $37°$ N. The number of subtropical tracers, passing through the central and western gates, are much smaller as compared with those passed the eastern ones. We distinguish two central gates V and III $134°$ E $-135.5°$ E and $132.5°$ E $-133.5°$ E, respectively, and the western gates I and II (Fig. 4c) in the range $130°$ E $-132.5°$ E. It follows from Fig. 4b that the western and central gates collect subtropical tracers mainly from the western part of the initial zonal line, from $129°$ E to $133°$ E. In other words,

water parcels from its eastern part ($133°$ E $-137°$ E) practically do not pass through those gates at the latitude $42°$ N. Thus, the western part of the initial material line at $37°$ N contributes to all the peaks at the tracer distribution $42°$ N, whereas its eastern part contributes mainly to the Tsushima peak.

To visualize the transport paths by which subtropical tracers reach the northern SF area we compute so-called tracking maps in Fig. 4S in Supplementary material showing where the subtropical tracers, which crossed eventually the zonal line $42°$ N,

wandered for the whole integration period.

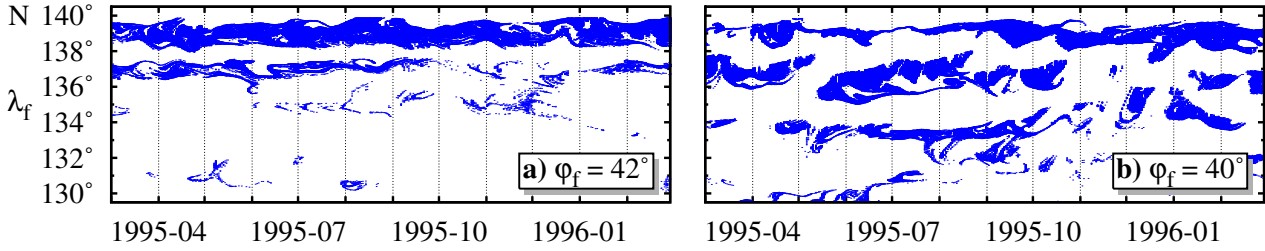

**Figure 5.** The $T - \lambda_f$ plots show when and at which longitudes the tracers, launched at the zonal line $37°$ N, crossed eventually the zonal lines a) $40°$ N and b) $42°$ N in the period from March 1, 1995 to March 1, 1996.

The $T - \lambda_f$ plots in Figs. 2S and 3S in Supplementary material show when and at which longitudes the tracers, launched weekly at the zonal line $37°$ N from January 2, 1993 to June 15, 2013, reached the zonal lines $40°$ N and $42°$ N, respectively. It is declared in Sec. 2 as the task 3. As an example, we show in Fig. 5 a typical $T - \lambda_f$ plot for the tracers crossed eventually the zonal lines $40°$ N and $42°$ N in the period from March 1, 1995 to March 1, 1996. It demonstrates the eastern gates VIII and

VII (Fig. 4) through which the subtropical tracers cross the corresponding latitudes. The locations of the central and western gates fluctuate in time, and some gates may be even closed for a while to the northward transport. The patchiness in the plot means that subtropical tracers prefer to cross the zonal lines in the specific places (note the peaks in Figs. 3) and during specific time intervals. Any patch with a large number of tracers somewhere, say, at the central meridional gate means that a water mass proportional to the size of this patch passed through the central gate across a given latitude during the period of

time proportional to its zonal size. Thus, the northward transport of subtropical water across the SF occurs by a portion-like





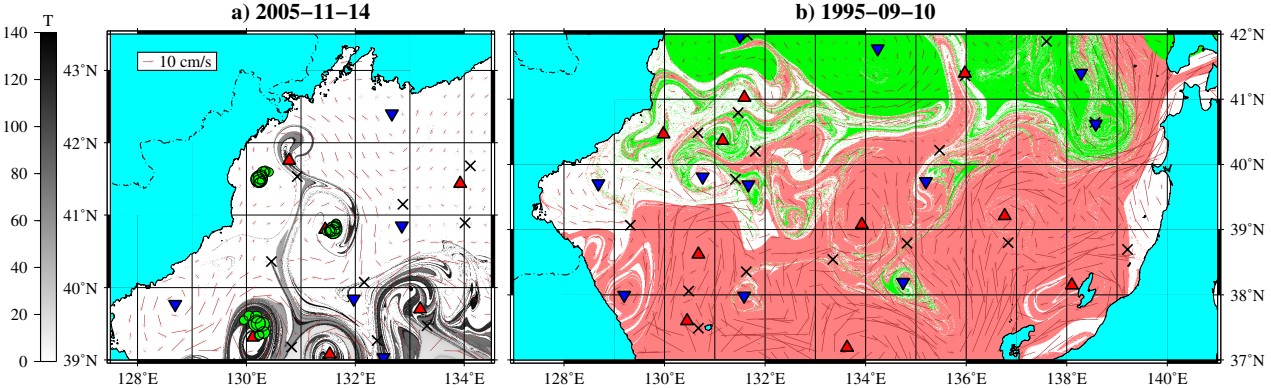

**Figure 6.** a) The Lagrangian map documents intrusions of subtropical water to the southern coast of Russia through the western gate. Nuances of the grey color code travelling time $T$ in days that took for subtropical tracers to reach their locations on the map from the latitude $37°$N to the dates shown. "White" tracers are those ones which did not come from the latitude $37°$N for the integration period, 140 days. Locations of available drifters are shown by full circles for one day before and after the dates indicated. b) The drift map documents a streamer-like northward transport of subtropical water across the front through a central gate with the help of the cyclone with the center at $41.5°$N, $134.4°$E. The AVISO velocity field is shown by arrows. "Instantaneous" elliptic and hyperbolic points, to be present in the area on a fixed day, are indicated by triangles and crosses, respectively. The red and green colors code the waters that entered the studied area for two years through its southern and northern boundaries, respectively. White color marks the tracers getting the coast.

manner. Specific oceanographic conditions may arise in a given area and at a given time which produce a large-scale intrusion of subtropical water to the north by means of mesoscale eddies to be present there.

One of the motivations of our work was an explanation of invasion of tropical and subtropical marine organisms in the northern part of the Sea, to the southern coast of Russia (Ivankova and Samuilov, 1979). To document an intrusion of subtropical

water there, we compute the backward-in-time Lagrangian maps (for a recent review of backward-in-time techniques see Prants, 2015). It is a realization of the task 4 in Sec. 2. The basin, shown in Fig. 6a, is seeded with a large number of tracers for each of which we compute the time required for a tracer to reach its location on the map to a fixed date from the latitude $37°$ N. The travelling time $T$ in days is coded by nuances of the grey color.

The map in Fig. 6a illustrates a mechanism of the penetration of subtropical water to the north through the western gate.

A vortex street with four anticyclones has been formed in the fall of 2005 to the north of the SF in the western part of the Sea. Their centers are marked in Fig. 6a by the elliptic points (triangles) with the coordinates $39.1°$ N, $131.5°$ E; $39.3°$ N, $130.1°$ E; $40.8°$ N, $131.4°$ E and $41.7°$ N, $130.8°$ E. Subtropical "grey" tracers propagate along the unstable manifolds of the three hyperbolic points between and around of those eddies to the north (simple description of the notion of stable and unstable manifolds in fluid flows can be found, e.g., in Prants, 2014). The hyperbolic points are marked by crosses in Fig. 6a with

the coordinates $39.2°$ N, $130.8°$ E; $40.3°$ N, $130.5°$ E and $41.6°$ N, $130.9°$ E. Thus, the vortex street provides an intrusion of subtropical water to the southern coast of Russia. The evidence of, at least, two anticyclones in the AVISO velocity field is





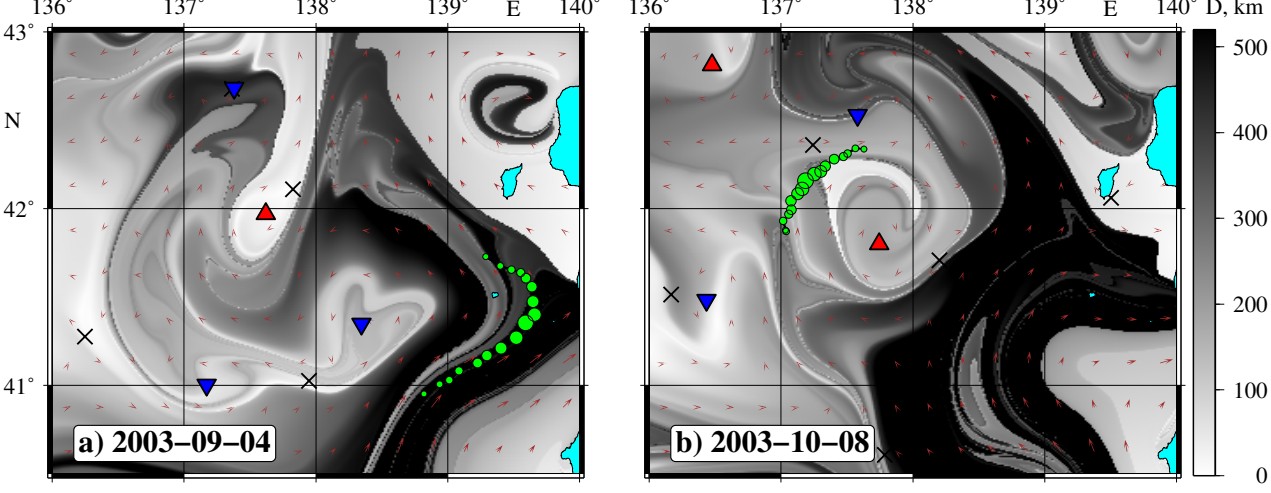

**Figure 7.** The drift maps with snapshots of the drifter's track superimposed show how the vortex pair facilitates transport of subtropical tracers to the northwest through the eastern gate. The displacement of particles $D$ in km, computed for a two months backward in time from the day indicated in each panel, are coded by shades of the grey color. Locations of the drifter No. 35660 are shown by full circles for two days before and after the day indicated.

confirmed by tracks of two available drifters (http://aoml.noaa.gov/phod/dac). Their locations are shown in Fig. 6a by full circles for one day before and after the date indicated on the map. The drifter No. 56739 has been trapped by the anticyclone with the center at 39.3° N, 130.1° E and the drifter No. 56746 — by the anticyclone with the center at 40.8° N, 131.4° E.

We have found such episodes with penetration of subtropical waters far to the north to the coast of Russia through the western gate in different years as well. Peripheries of mesoscale eddies in the ocean are known to be transport pathways for larvae, fish and other marine organisms. In our case they might be a kind of transport for heat-loving organisms to reach the southern coast of Russia (Ivankova and Samuilov, 1979).

An example of the intrusion of subtropical water through the central gate across the SF is shown in Fig. 6b with another kind of Lagrangian maps, so-called backward-in-time drift maps (Prants et al., 2011a, 2014a) computed as part of the task 4. In

the beginning of September, 1995 a mesoscale cyclonic eddy to the north of the SF with the center at about 41.5° N, 134.4° E "grabbed" some subtropical water at its southern periphery and pulled it to the north. The red and green colors on backward-in-time drift maps code the waters that entered the studied area for two years through its southern and northern boundaries, respectively. In the course of time the streamer-like intrusion of subtropical tracers reached the latitude 42° N moving to the north (Fig. 6b).

As to the transport of subtropical waters through the eastern gate VII (see Fig. 3 and Fig. 4), it occurs mainly due to existence of a quasi-permanent vortex pair labelled as AC-C in the mean field in Fig. 1. It provides a propulsion of some subtropical tracers to the northwest whereas most of them, propagating along the eastward frontal jet, join to the Tsushima Warm Current

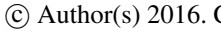


**Nonlinear Processes**
**in Geophysics**
Discussions

and flows out to the Pacific through the Tsugaru Strait. The maps in Supplementary material (Figs. 5S and 6S) document a typical situation with a propulsion of subtropical water to the northwest in September – October, 2003. The browsing and analysis of Lagrangian drift maps, computed for the whole observation period, have shown that frontal eddies used to facilitate the northward transport of subtropical water across the SF via the central and eastern gates.

To illustrate how this quasi-permanent vortex pair works we show in Fig. 7 the drift map for tracers distributed over the area and advected for two months backward in time starting from the dates indicated. The values of displacements of the tracers, $D$, in km are coded by shades of the grey color. So, the black tracers have displaced for the same time considerably as compared to the white ones. To verify our simulation we show in Fig. 7 positions of the drifter No. 35660 by full circles for two days before and after the date indicated with their size increasing in time. The entire track of that drifter, launched on May 2, 2003

at the point $34.925°$ N, $129.3°$ E, is shown in Fig. 7S in Supplementary material.

     In the beginning of September, 2003 (Fig. 7a) the vortex pair at the entrance to the gate VII consists of an anticyclone with the center at about $42°$ N, $137.7°$ E and a cyclone $41.25°$ N, $138.35°$ E. The cyclone winds some subtropical water from the eastward frontal jet round its northern periphery in a streamer-like manner (see the black tongue in Fig. 7a). Then this water is wound by the anticyclone round its southern periphery and is propulsed to the northwest. It is confirmed by snapshots of the

track of the drifter No. 35660 for September – October, 2003 (see Fig. 6S in Supplementary material). Being in the beginning of September in the main stream (Fig. 7a), it has drifted round the cyclone for the first half of September, then round the anticyclone for the second half of September and in the beginning of October. Eventually the drifter No. 35660 crossed the latitude $42°$ N (Fig. 7b) and moved to the north lugged by modified subtropical waters.

### 3.3    Discussion of the effect of possible altimetry errors on statistical features of Lagrangian transport

It has been shown statistically in this section that the average northward component of the AVISO velocity field dictates preferred near-surface transport pathways of subtropical waters in the central JS. The ability of satellite altimetry to accurately measure sea level anomalies has vastly improved over the last decade. However, there are still some measurement errors due to different reasons that lead to errors in the velocity field provided by the AVISO we used.

     Some simulation results with an imperfect AVISO velocity field can be verified by comparing them with satellite, drifter and

*in situ* observations. It has been done in this paper when possible. The AVISO velocity field, averaged for the whole observation period (Fig. 1), reproduces all the known main mesoscopic features of near surface circulation in the Sea (Hirose et al., 2005; Lee and Niiler, 2005; Danchenkov et al., 2006; Talley et al., 2006; Yoon and Kim, 2009; Kim and Yoon, 2010; Lee and Niiler, 2010; Ito et al., 2014) including not only pathways of the main currents but even locations of Ulleung, Dok, Oki, Wonsan and other quasi-permanent mesoscale eddies (Fig. 1). Moreover, the simulation has been compared with available tracks of drifters

(see Figs. 3, 6 and 7).

     In this section we discuss possible effect of errors in the altimetry field on our simulation results. Nobody knows, of course, "true" velocity field in the real ocean. The AVISO velocity field has errors as compared with an unknown "true" velocity field which can be considered as a noise $\Delta(u,v)$. The question is how reliable are our statistical simulation results based on an imperfect AVISO velocity field? To which extent one can trust to them? All the simulation results, based on the average





AVISO velocity as in Fig. 1, are supposed to be reliable because the errors are averaged out for 22 years. As to other simulation results, they depend on possible noise $\Delta v$ in the AVISO northward component $v_+$ which could, in principle, change the results but only if the noise would be strong enough to change direction of the meridional velocity, i.e., if $\Delta v > |v|$. If the average AVISO northward component $v_+$ is large enough as in the areas with dominated northward currents, we don't expect that it

would be changed there significantly under influence of noise. So, locations of the preferred transport pathways in Figs. 3 and Fig. 4, which are dictated by that component, are not expected to be changed significantly.

If the average AVISO northward component $v_+$ is small, then two options are possible.

1) It is small due to domination of a southward current somewhere, i.e., $v_- \gg \Delta v$. It is clear that possible noise has practically no effect on northward transport in this case. Say, the forbidden zone in Fig. 2a to the south off Vladivostok, where

northward transport has not been observed during the whole observation period, should be located there at any realistic level of noise because it exists due to domination of a sufficiently strong southward jet (VMJ in Fig. 1).

2) The average AVISO northward component $v_+$ is small due to a smallness of the absolute velocity, i.e., $\sqrt{u^2 + v^2} \sim \Delta v$. In this case northward and southward transports are equalized, and they are small if the noise is small enough. It is hardly to expect such a situation along the SF because of a plenty of mesoscale eddies along the front where the absolute velocities are

not small.

As to influence of possible errors in altimetry-derived velocity field on concrete mesoscale features, it has been studied by Harrison and Glatzmaier (2012); Hernández-Carrasco et al. (2011); Keating et al. (2011) how an additional noise in the advection equations, modelling unknown corrections to the AVISO velocity field, might change Lagrangian coherent structures revealed by the finite-time Lyapunov technique. Strongly attracting and repelling individual Lagrangian coherent structures

in the California Current System have been shown to be robust to perturbations of the velocity field of over 20% of the maximal regional velocity (Harrison and Glatzmaier, 2012). Individual trajectories have been shown to be sensitive to small and moderate noisy variations in the velocity field but statistical characteristics and large-scale structures like mesoscale eddies and jets are not (Cotte et al., 2010; Hernández-Carrasco et al., 2011; Keating et al., 2011).

## 4 Conclusions

The main results of altimetry-based simulation and analysis of the northward near-surface Lagrangian transport of subtropical water across the Japan Sea frontal zone for the period from January 2, 1993 to June 15, 2015 are the following.

1. A methodology to simulate and analyze Lagrangian large-scale transport in frontal areas is developed (tasks 1–4 in Sec. 2).

2. There are "forbidden" zones in the Japan Sea where the northward transport has not been found during all the observation

period (see the rectangles in Fig. 2a). The "forbidden" zone to the south off Vladivostok exists due to a quasi-permanent southward jet there (see VMJ in Fig. 1). The other "forbidden" zone exists due to the presence of a quasi-permanent



topographically constrained anticyclonic eddy with the center at about $41.3°$ N, $134°$ E in the deep Japan Basin and the eastward zonal jet blocking northward transport there (see AC in Fig. 1).

3. Northward near-surface Lagrangian transport of subtropical water across the Subpolar Front has been statistically shown to be meridionally inhomogeneous with specific gates and barriers in the frontal zone whose locations are determined by the local advection velocity field (see pronounced peaks in Figs. 3 and Fig. 4).

4. The transport through the gates has been shown to occur by a portion-like manner, i.e., those gates "open" during specific time intervals (see a patchiness in Fig. 5 and Figs. 2S and 3S).

5. The gates "open" due to suitable dispositions of mesoscale frontal eddies facilitating propagation of subtropical waters to the north. It is documented for the western, central and eastern gates with the help of different kinds of Lagrangian maps and validated by some tracks of available drifters (see intrusions of subtropical tracers around the eddies in Figs. 6, 7, 5S and 6S). In particular, invasion of tropical and subtropical marine organisms in the northern part of the Sea, to the southern coast of Russia, can be explained by the presence of vortex streets at the western gate (Fig. 6).

*Acknowledgements.* This work was supported by the Russian Science Foundation (project No. 16–17–10025). A publication cost is covered, in part, by the Office of Naval Research Grant No. N00014-16-1-2492. The altimeter products were distributed by AVISO with support from CNES.

Supplementary materials associated with this paper can be found in the on-line version.



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
