# Peer review of "Statistical analysis of Lagrangian transport of subtropical waters in the Japan Sea based on AVISO altimetry data"

_Nonlinear Processes in Geophysics, 2016_

## Referee Comment (RC1) · Anonymous Referee #1 · 14 Nov 2016

**Review for the «Nonlinear Processes in Geophysics. Discussions» manuscript entitled «Statistical analysis of Lagrangian transport of subtropical waters in the Japan Sea based on AVISO altimetry data» by S.V. Prants, M.V. Budyansky, and M.Yu. Uleysky**

The problem of cross-frontal transport is one of the most important in the dynamics of the atmosphere, the oceans and seas. In this paper, the authors approach this problem on the basis of altimetry-based simulation and analysis of Lagrangian transport in relation to the subtropical water across the Japan Sea frontal zone. Especially interesting are the results of the establishment of the "open gate" and "forbidden zones", as well as defining of theirs regimes.

Note, that the circulation of the Japan Sea was an object of study by many authors for a long time. In my opinion, this study lacked a comparison with the schemes of currents obtained, for example, in

- Chang, K.-I., Teague, S.J., Lyu, S. J., Perkins, H. T., Lee, D.-K., Watts, D. R., Kim, Y.-B., Mitchell, D.A., Lee, C.M., and Kim, K. Circulation and currents in the southwestern East/Japan Sea: Overview and review. Progr. Oceanogr., 61, 105-156, doi:10.1016/j.pocean.2004.06.005, 2004.
- Holloway, G., Soul, T., and Eby, M. Dynamics of circulation of the Japan Sea. J. Mar. Res., 53, 539-569, doi:10.1357/0022240953213106, 1995.
- Kawabe, M. Branching of the Tshushima Current in the Japan Sea. Part I. Data analysis. J. Oceaonogr. Soc. Japan, 38, 95-107, doi:10.1007/BF02110295, 1982.
- Kawabe, M. Branching of the Tshushima Current in the Japan Sea. Part II. Numerical experiment. J. Oceaonogr. Soc. Japan, 38, 183-192, doi:10.1007/BF02111101, 1982.
- Sekine, Y. Wind-driven circulation in the Japan Sea and its influence on the branching of the Tsushima Current. Progr. Oceanogr., 17, 297-313, doi:10.1016/0079-6611(86)90051-0, 1986.
- Senjyu, T. The Japan Sea intermediate water; its characteristics and circulation. J. Oceanogr. 55, 111-122, doi:10.1023/A:1007825609622, 1999.
- Takano, K., ed. 1991. Oceanography of Asian Marginal Seas, Elsevier, 431 pp.
- Yoon, J. H. Numerical experiment on the circulation in the Japan Sea. Part I: Formation of the East Korea Warm Current. J. Oceanogr. Soc. Japan, 38, 43-51, doi:10.1007/BF02110289, 1982.
- Yoon, J. H. Numerical experiment on the circulation in the Japan Sea. Part II: Influence of seasonal variations in atmospheric conditions on the Tsushima Current. J. Oceanogr. Soc. Japan, 38, 81-94, doi: 10.1007/BF02110294, 1982.
- Yoon, J. H. Numerical experiment on the circulation in the Japan Sea. Part III: Formation of the nearshore branch of the Tsushima Current. J. Oceanogr. Soc. Japan, 38, 125-130, doi:10.1007/BF02110283, 1982.
- You, Y., Chang, K.-I., Yun, J.-Y., and Kim, K.-R. Thermocline circulation and ventilation of the East/Japan Sea, part I: Water-mass characteristics and transports. Deep Sea Res. Part II: Topical Studies in Oceanography, 57, 1221–1246, doi:10.1016/j.dsr2.2009.12.011, 2010.

This work, of course, is of interest to potential readers of NPG. The paper should be published after taking into account this deficiency.

---

## Referee Comment (RC2) · Anonymous Referee #2 · 28 Nov 2016

In this paper, the authors made a detailed Lagrangian analysis of currents in the Japan Sea for 21 years of AVISO data. An interesting result is the location of "gates" and "barriers" to the transport. The position and timing of these gates is analyzed in detail in terms of the advection currents that affect the Japan Sea.

The paper merits for publication in NPG, however some minor changes should be done that in my opinion may enrich the text.

Colors or gray shades in Fig.2 should be different as it is impossible to see anything.

I do not understand why to compare with real drifters in Fig.3. Drifters and tracers do not match as it is already stated by the authors. The corresponding blue line should be

deleted from Fig.3. Vertical black lines in Fig.3 confuse the reader.

Why do you choose so strange latitudes as 39.875, 40.125, etc in Fig.3 when in the rest of the paper, latitudes are integer numbers as 40N or 42N? Could you re-draw that Figure to be consistent?

Besides, Fig.4 is quite obscure, and the results shown there are already shown in Fig.3. Why do not define the regions (I to VIII) in Fig.3? Clearly, due to the coast line and currents most of the particles should move towards the east as they move northward as it is already shown in Fig.3 and repeated in Fig.4. In my opinion Fig.4 could be deleted.

An interesting result shown in Fig.5 is that gates may be closed during some period of time (white patches at certain longitudes). However the reason for that it is not clear for me. May be the authors should make an effort to explain that more clearly. These patches repeat regularly on time? may be with a seasonal period?

Finally, the English style should be checked, but I suppose the journal will take care of that later on.

---

## Referee Comment (RC3) · Anonymous Referee #3 · 30 Nov 2016

This manuscript presents a statistical analysis of the transport processes from geostrophic velocities obtained from the AVISO SLA product. In particular the authors use Lagrangian techniques to study the northward transport in the Japan Sea of surface water masses coming from subtropical regions. This study is conducted combining trajectories of virtual particles, advected by the altimetry velocity field, and real drifters trajectories.

Some of the findings are: spatial pattern of time averages of particles reaching northern regions shows zonal dynamical features at middle latitudes that are candidates to act as gates/barriers to transport; there are small regions in the north where particles coming from the south does not arrive over the analyzed time period ("forbidden

zones"); in general, particles coming from the western part are transported to the eastern; there are some intrusions of subtropical particles near the Russian coast in the northwest of the Japan sea basin; mechanism to explain the dominant northwestward transport of particles is given in relation to the presence of a quasi-permanent frontal pair-vortex system; adding errors in the altimetry velocity data do not affect significantly the results.

In general, the authors present a interesting work aimed at improving the description of surface transport in the Japan Sea in terms of Lagrangian statistic indicators. This is a good piece of work which can be of interest to some NPG readers. However a minor/major revision has to be addressed before publication. The main issues that need to be clarified by the authors are listed bellow.

Major concerns: General: 1. Although the study has relevant results to understand the mesoscale transport dynamics at the surface of the Japan Sea, I found that some of the results are repeated throughout the manuscript, becoming redundant. For instance the information that can be obtained from Figure 4 can be deduced from the other figures. 2. Also it's really hard to follow the storyline of the study, I have found many unnecessary repetitions of descriptions and a lot of sentence and expressions that do not make sense. The manuscript needs a careful read through. A copy editing of the text would be highly appreciated to make the reading easier. 3. The authors analyze how is the northward transport in relation to zonal fronts located at the middle of the basin of the Japan Sea near 40°N. However looking at Figure 1 one can see that these fronts extends from 37°N to 42°N. In fact the authors only analyze the flux of particles at latitudes from 39.875 to 42.125 (Figure 3). Please could you clarify this choice? 4. Have the authors compared the intrusions of subtropical waters (figure 6) with Sea Surface Temperature or Ocean color satellite images? It could help you to prove with observations such dynamical features?

Abstract: 1. The authors repeat several times "Lagrangian indicators (line 2)", Lagrangian maps (line 7) without specifying the Lagrangian diagnosis used in the study.

3. Sentence in lines 5-6. It is hard to understand how the eddies open the gates due to their suitable dispositions. What does "suitable dispositions" means?

2. Last two sentences: It's clear that something is happening because of the peculiarities of the advection velocity field, but which one? Please be more precise.

Introduction:

1. Figure 1 shows high values of northeastward velocity data in the TS (Tsugaru strait)? It means that, in average, the most of particles scape throughout this strait to the Pacific Ocean? Do the authors know if there is a realtionship between these feature with the frontal-pair-vortex systems (AC-C) located at the entrance of the strait?

2. Please move to the first paragraph (where you are describing the bathymetry) the sentence in line 19 (page 1) with the reference to the figure showing the bathymetry.

3. Acronyms: The text is full of acronyms. - Could the authors explain what does SF mean (line 5, page 2)? Salinity fronts? - It could be great if the authors describe all the acronyms in the introductory part (second paragragh of introduction section) and not in the Results section (first paragraph of section 3.1) neither in the caption of Figure 1.

Data and methods:

1. The time step used for the Runge-Kutta integration is 1/1000. It means 0.001 days ($\sim$1.5 minutes)? Why this such short time step? Have you compared the results using a longer time step, namely, 1 day (the time resolution of the Altimetry data?)

2. It is hard to understand the sentences from line 18 and line 21 page 4: " Trajectory of each . . .........in the northward transport only." Authors say that t"he fixed the position and time of each tracers when they cross a given latitude between 37 and 43" and then they say that they "fix only the first crossing of a given latitude". Could the authors clarify this part of the methodology?

3. line 21 page 4. Could the authors explain how a cell of AVISO can have two corners

situated at land? As far as I am concerned a cell of AVISO only can be either land or ocean depending of the land mask. Maybe they mean two corners of the integration cell situated at land?

4. line 23-25. Could the authors read carefully and rewrite the paragraph? The sentence does not make sense.

5. Could the authors provide a description of the methodology used to compute and classify the fixed points of the velocity field in elliptic and hyperbolic points? The velocity field in the centers of the eddies is different to zero. Could the streamlines of saddle nodes be used to localize fronts?

6. Please specify the number of drifters used in the study?

Results and Discussion

1. The description given in the first paragraph is already provided in the introductory part (line 31 page 3 and lines 1, 2 page 4).

2. First and second paragraph of this section are not results and they should be in the Introduction or in the Data and methods sections.

3. Second paragraph page 7. Eulerian (time average of northward velocities, Fig. 2b) with Lagrangian (particles trajectories crossing different latitudes, Fig. 2a) diagnosis are compared. Authors state that both diagnosis are equivalent because the transport is determined by local advection. It means that the dynamics is controlled by the local scales ("small scales") and not by the large scales?

4.Do Fig. 3 show zonal cross-sections of Fig. 2 for four latitudes? If so, why the authors do not relate both figures in the manuscript, it could help to the reader. I do not know what does means "central part"? Authors state that "the correlation is rather good for western and eastern parts but not for the central one" (line 31-32, page 7), however, I see a good correlation between virtual and real drifters in the central part of the Japan Sea (see minima in Fig. 3 at 134°E-136°E). Moreover, this correlation is stronger in

the north part (higher latitudes) than in the southern. Maybe because of the number of available drifters?. Could the authors specify the number of drifters? Have the authors values of the correlations coefficients?

5. With regard to the bad correlation with drifters (line 32-33 page 7), other explanation could be because drifters movement is due also to submesoscale features which are not provided by velocities derived from altimetry.

6. Line 2 page 8. How does the authors know that drifters have been launched randomly over the basin? Could they provide references on these drifters experiments explaining these random releases?

6. Drifters are not launched at 37°N but at least they have to cross this latitude to take them into account in the meridional transport computations.

7. Second paragraph is confused. I do not know how to differentiate gates and barriers from Fig 3. Minima and maxima of numbers of tracers?. Could the authors clarify this point?

8. It is difficult to deduce from Figure 4 the parameter chosen by the authors to define the size of gates and barriers.

9. Line 1-2, page 11. It's hard to understand what the authors mean in this sentence.

10. Line 3, page 11- This motivation has been included in the introductory sections. Please avoid repetitions.

11. Line 6, page 11. Definition of Lagrangian Maps is not clear: I really do suggest that the authors rewrite it.

12. Fig 6 correspond to maps of residence times computed backward in time. They are the incoming times and it has been already used for several authors:

- Lipphardt, B., Jr., Small, D., Kirwan, A., Jr., Wiggins, S., Ide, K., Grosch, C., and Paduan, J.: Synoptic Lagrangian maps: Aplica tion to surface transport in Monterey

Bay, J. Mar. Res., 64, 221– 247, 2006.

- H. Gildor, E. Fredj, J. Steinbuck, S. Monismith. Evidence for submesoscale barriers to horizontal mixing in the ocean from current measurements and aerial photographs. Journal of Physical Oceanography, 39, 1975–1983, 2009.

- Hernández-Carrasco, I., López,C., Orfila, A., and Hernández-García, E.; Lagrangian transport in a microtidal coastal area: the Bay of Palma, island of Mallorca, Spain. Nonlin. Processes Geophys., 20, 921–933, 2013.

13. It could be great if authors include a reference when they say that Peripheries of the mesoscale eddies in the ocean are known to be transport pathways for larvae, fish . . ...... (lines 5-6, page 12).

14. Sentence in line 7: ". . ...kind of transport for heat-loving organisms to reach the southern coast of Russia". It means that the coast of Russia is warm?

15. The paragraph (lines 8-15) could be improved if the authors reorganize the text. Lines 11-13 (" The red and green . . ......respectively") could be move in before line 10 ("In the beginning of September . . ....pulled to the north.")

16. Line 3, page 13. It seems that this is the mechanism to explain the intrusions of subtropical waters (Figure 6). The manuscript could be improved if authors are more explicits.

17. Is the displacements of the tracers (D) a Lagrangian diagnosis developed by other authors, by means of the function M, in order to obtain the Lagrangian Coherent Structures?:

- Mendoza, C. and Mancho, A. M.: The hidden geometry of ocean flows, Phys. Rev. Lett., 105, 038501, doi:10.1103/PhysRevLett.105.038501, 2010.

18. Line 7, page 13. " so, the black tracers . . .......white ones" I do not understand what the authors mean with "displaced". What is the difference between white and black

colors? The length of the trajectory?

19. Line 14, page 13: "....pro-pulsed to the northwest". Do the authors mean "northeast" instead of "northwest"? It seems that particles are not transported northwestward by the frontal-pair-vortex system but northeastward.

20. I do not understand the last paragraph from line 14 to line 18, page 13. Are the authors describing the drifter trajectories?

21. Line 24: Simulations with "imperfect" AVISO is compared with satellite, drifter and in situ observations. What kind of satellite product has been used to be compared with AVISO? When comparing with drifter trajectories you should take into account that drifter velocities has to be different than the geostrophic velocities derived from AVISO by definition.

22. Line 25-30. This sentence has been mentioned three times in the manuscript. Please avoid repetitions.

23. Line 31: "Nobody knows, of course true velocity field in the real ocean" This sentence is not necessary, please remove it.

24. Line 32: "The AVISO velocity field has errors as compared with an unknown "true" velocity field...". This sentence is not logical: in my opinion anything can not be compared with things that are unknown, because they are unknown. One could say that the unknown part of the velocity field could be simulated by adding noise in the velocity data.

25. Line 34, page 13. What does "To which extent one can trust to them" mean?

26. Line 19, page 4. This study of adding errors in the velocity field to compute the trajectories has been performed for the FTLE and also for FSLE.

Conclusions

1. I have found many repetitions that have been copied from the body of the

manuscript. I suggest that the authors rewrite the whole conclusions section.

2. Please remove references to the Figures.

Technical comments:

1. Please remove "there" (last word of abstract, line 11, page 1)

2. Why "Sea" (line 5 page 2) is in capital letter?

3. Line 6 page: 3 replace "to the coast" with "in the coast"

4. Line 10, page 3: Remove "across it".

5. Line 11 page 3: in " we use the altimetry data" remove "the".

6. Line 14 page 3. Remove "based on altimetry data". It has already been stated several times in the text. Please avoid repetitions.

7. Line 25, 26, page 7, It is not necessary to specify the longitudes for the zonal cross-section but only with the latitude the reader can localize the zonal lines.

8. Please read carefully the lines 28 in page 7 and rewrite the text: " The number of crossing . . .....dashed curves" by for instance "The number of available drifters crossing the given latitudes are shown . . ........".

9. There are many "to the" through the manuscript that should be replaced with "at the" or "in the"

10. Line 1 , page 9. What does "task 2" mean? Authors use "task X" throughout the manuscript to refer to the computations given in the Data and methods section. It could be great if it is stated in the methodology section that the computation X correspond to task X. (Although task is usually used in the context of a project and not in a paper).

11. There are two reference for Prants et al, 2015.

12. Line 1, page 12. Please remove the website of drifter data (it has been shown in

Data and Methods) and also the number of the drifters.

13. Line 23, page 13. Remove "the" before AVISO and "we used" after AVISO.

14. Line 5, page 14. Please replace the sentence " So , locations of the preferred . . ....are possible" with "So, locations of the preferred transport pathways are not expected to be changed significantly".

15. Line 9, page 14. Please replace "Say" with "For example".

16. Line 14, page 14. Replace "a plenty of " by "the presence of numerous".

17. Line 16, page 14. Replace "As to" by "The" and remove "it"

18. Line 17, page 14. Add "by analyzing" just before "how and additional".

19. Line 18, page 14. remove "modelling unknown corrections to the AVISO velocity field". It has already been mentioned.

20. Line 19, page 14. Add "Finite Size Lyapunov Exponents" after "Finite Time Lyapunov Exponents"

Please also note the supplement to this comment:
http://www.nonlin-processes-geophys-discuss.net/npg-2016-67/npg-2016-67-RC3-supplement.pdf

---

## Author Comment (AC1) · 23 Dec 2016

The author's respond to the Reviewer #1

Reviewer #1. Recommendation: The paper should be published after taking into account this deficiency.

Reviewer's comments Note, that the circulation of the Japan Sea was an object of study by many authors for a long time. In my opinion, this study lacked a comparison with the schemes of currents obtained, for example, in • Chang, K.-I., Teague, S.J., Lyu, S. J., Perkins, H. T., Lee, D.-K., Watts, D. R., Kim, Y.-B., Mitchell, D.A., Lee, C.M., and Kim, K. Circulation and currents in the southwestern East/Japan Sea: Overview and

review. Progr. Oceanogr., 61, 105-156, doi:10.1016/j.pocean.2004.06.005, 2004. • Holloway, G., Soul, T., and Eby, M. Dynamics of circulation of the Japan Sea. J. Mar. Res., 53, 539-569, doi:10.1357/0022240953213106, 1995. • Kawabe, M. Branching of the Tshushima Current in the Japan Sea. Part I. Data analysis. J. Oceaonogr. Soc. Japan, 38, 95-107, doi:10.1007/BF02110295, 1982. • Kawabe, M. Branching of the Tshushima Current in the Japan Sea. Part II. Numerical experiment. J. Oceaonogr. Soc. Japan, 38, 183-192, doi:10.1007/BF02111101, 1982. • Sekine, Y. Wind-driven circulation in the Japan Sea and its influence on the branching of the Tsushima Current. Progr. Oceanogr., 17, 297-313, doi:10.1016/0079-6611(86)90051-0, 1986. • Senjyu, T. The Japan Sea intermediate water; its characteristics and circulation. J. Oceanogr. 55, 111-122, doi:10.1023/A:1007825609622, 1999. • Takano, K., ed. 1991. Oceanography of Asian Marginal Seas, Elsevier, 431 pp. • Yoon, J. H. Numerical experiment on the circulation in the Japan Sea. Part I: Formation of the East Korea Warm Current. J. Oceanogr. Soc. Japan, 38, 43-51, doi:10.1007/BF02110289, 1982. • Yoon, J. H. Numerical experiment on the circulation in the Japan Sea. Part II: Influence of seasonal variations in atmospheric conditions on the Tsushima Current. J. Oceanogr. Soc. Japan, 38, 81-94, doi: 10.1007/BF02110294, 1982. • Yoon, J. H. Numerical experiment on the circulation in the Japan Sea. Part III: Formation of the nearshore branch of the Tsushima Current. J. Oceanogr. Soc. Japan, 38, 125-130, doi:10.1007/BF02110283, 1982. • You, Y., Chang, K.-I., Yun, J.-Y., and Kim, K.-R. Thermocline circulation and ventilation of the East/Japan Sea, part I: Water-mass characteristics and transports. Deep Sea Res. Part II: Topical Studies in Oceanography, 57, 1221–1246, doi:10.1016/j.dsr2.2009.12.011, 2010.

Our respond We thank the Reviewer for providing us the above mentioned references. We compared briefly in the revised text our averaged AVISO velocity field and its main large-scale features, including persistent mesoscale eddies in the basin, with the schemes of the Japan Sea (JS) surface circulation obtained in the references proposed by the Reviewer. We referred to most of those references in Introduction and in the first paragraph in Sec.3.1 in addition to already referred papers on this subject.

[Figure]

However, the main intent of our ms was not a review of currents in the JS and their variability but a Lagrangian statistical analysis of near-surface transport of subtropical waters in the JS frontal area.

Please also note the supplement to this comment:
http://www.nonlin-processes-geophys-discuss.net/npg-2016-67/npg-2016-67-AC1-supplement.pdf

---

## Author Comment (AC2) · 23 Dec 2016

The author's respond to the Reviewer #2

Reviewer #2. Recommendation: The paper merits for publication in NPG, however some minor changes should be done that in my opinion may enrich the text.

1. Cited from the referee's report Colors or gray shades in Fig.2 should be different as it is impossible to see anything. I do not understand why to compare with real drifters in Fig.3. Drifters and tracers do not match as it is already stated by the authors. The corresponding blue line should be deleted from Fig.3. Vertical black lines in Fig.3 confuse the reader. Our response As to Fig.2, it does not pretend to give an exact quantitative

information. It designed to show a meridional inhomogeneity of northward transport of subtropical water across the Subpolar Front with an increased (decreased) density of points corresponding to transport gates and barriers, respectively. It is clear that areas with increased density of points in Fig. 2a correlate well with areas with increased average values of the northward component of the AVISO velocity field in Fig. 2b. The chosen color gradation has been found empirically to be optimal to represent forbidden zones as bright spots. We would like to stress that the density difference in some meridional ranges in Fig. 2a may be very large because of the logarithmic-scale representation. As to Fig.3, the meridional dependence of the number of crossings of zonal lines by available drifters confirms partly the existence of simulated forbidden zones. The vertical black lines in Fig.3 are plotted to compare different curves and their maxima and minima. We rewritten the first two paragraphs in Sec.3.2 to be: "Now let's look more carefully at the meridional distribution of subtropical tracers crossed the Subpolar Front for the whole period of simulation. We choose for reference four zonal lines along the AVISO grid at \N{42.125}, \N{41.875}, \N{40.125} and \N{39.875}. They are shown in Fig.~\ref{fig3} by solid curves with superimposed meridional distributions of the averaged northward AVISO velocity (arrows). The number of crossings of those latitudes by available 333 drifters is shown by dashed curves. The correspondence between the peaks in the meridional distributions of the tracers, drifters and the averaged northward AVISO velocity is rather good for all the chosen zonal lines confirming their direct connection. However, the comparison with drifters should be taken with care because of a comparatively small number of available drifters. Drifters, are not ideal passive tracers, and their motion is subjected to submesoscale features which were not caught by altimetry-derived data. Moreover, the drifters have not been launched at the zonal line \N{37} like artificial tracers in simulation. Their launch sites for more than 20 years have been distributed rather randomly over the basin." The correspondence between the curves for drifters and the averaged northward component of the AVISO velocity is rather good at 40 N. It is worse at 42 N, however, it is clear that the drifters are transported mainly by the Tsushima Current. The vertical black lines in

Fig.3 facilitate a comparison between different curves.

2. Cited from the referee's report Why do you choose so strange latitudes as 39.875, 40.125, etc in Fig.3 when in the rest of the paper, latitudes are integer numbers as 40N or 42N? Could you re-draw that Figure to be consistent? Our response In Fig.3 they are chosen to fit the AVISO grid where we estimate the AVISO velocities. We need not that in other figures. 3. Cited from the referee's report Besides, Fig.4 is quite obscure, and the results shown there are already shown in Fig.3. Why do not define the regions (I to VIII) in Fig.3? Clearly, due to the coast line and currents most of the particles should move towards the east as they move northward as it is already shown in Fig.3 and repeated in Fig.4. In my opinion Fig.4 could be deleted. Our response Figure 4 contains new information as compared to Fig.3. It shows how many and at which final longitudes $\lambda\_f$ the tracers with initial longitudes $\lambda\_0$, launched weekly at t 37N from January 2, 1993 to June 15, 2013, were able to cross 40N and 42N latitudes. The three paragraphs in Sec.3.2 are devoted to describe this new information.

4. Cited from the referee's report An interesting result shown in Fig.5 is that gates may be closed during some period of time (white patches at certain longitudes). However the reason for that it is not clear for me. May be the authors should make an effort to explain that more clearly. These patches repeat regularly on time? may be with a seasonal period? Our response It is defined by the properties of the local advective velocity field. The gates open due to suitable dispositions of mesoscale frontal eddies facilitating propagation of subtropical waters to the north. It is documented in the paper with the help of different kinds of Lagrangian maps in Figs.6 and 7 and validated by tracks of available drifters. For example, a vortex street with four anticyclones has been formed in the fall of 2005 to the north of the Subpolar Front in the western part of the sea. Their centers are marked in Fig. 6a by the elliptic points (triangles) with the coordinates 39.1N, 131.5 E; 39.3N, 130.1E; 40.8N, 131.4E and 41.7N,130.8E. It is explained on p.11 (L.10-15), on p.12 (L.80-15) and on p.11 (L.10-15) in the text. The gates open at those places where suitable dispositions of eddies and vortex streets
Interactive comment

appear time to time to facilitate the northward transport. They may appear due to different reasons, e.g., due to a seasonal variability of the current field, migration of frontal eddies, etc.

5. Cited from the referee's report Finally, the English style should be checked, but I suppose the journal will take care of that later on. Our response We did our best to edit all the text.

Please also note the supplement to this comment:
http://www.nonlin-processes-geophys-discuss.net/npg-2016-67/npg-2016-67-AC2-supplement.pdf

---

## Author Comment (AC3) · 23 Dec 2016

The author's respond to the Reviewer #3

We are very grateful to the Referee for a very careful reading of the manuscript and a number of useful comments and critics we tried to take into account in the revised version.

III. Reviewer #3: Recommendation - In general, the authors present a interesting work aimed at improving the description of surface transport in the Japan Sea in terms of Lagrangian statistic indicators. This is a good piece of work which can be of interest to some NPG readers. However a minor/major revision has to be addressed before

publication. The main issues that need to be clarified by the authors are listed bellow.

Responses to the Third Reviewer's report Major concerns: GENERAL REMARKS: 1. Cited from the referee's report 1. Although the study has relevant results to understand the mesoscale transport dynamics at the surface of the Japan Sea, I found that some of the results are repeated throughout the manuscript, becoming redundant. For instance the information that can be obtained from Figure 4 can be deduced from the other figures. Our response Figure 4 shows how many and at which final longitudes $\lambda\_f$ the tracers with initial longitudes $\lambda\_0$, launched weekly at 37N from January 2, 1993 to June 15, 2013, were able to cross 40N and 42N latitudes. The three paragraphs in Sec.3.2 are devoted to describe this new information. This information is absent in other figures in the paper.

2. Cited from the referee's report 2. Also it's really hard to follow the storyline of the study, I have found many unnecessary repetitions of descriptions and a lot of sentence and expressions that do not make sense. The manuscript needs a careful read through. A copy editing of the text would be highly appreciated to make the reading easier. Our response We did our best to improve the English, to edit the paper, to remove repetitions and to clarify the sentences and expressions indicated by the referee.

3. Cited from the referee's report 3. The authors analyze how is the northward transport in relation to zonal fronts located at the middle of the basin of the Japan Sea near 40 N. However looking at Figure 1 one can see that these fronts extends from 37N to 42 N. In fact the authors only analyze the flux of particles at latitudes from 39.875 to 42.125 (Figure 3). Please could you clarify this choice? Our response Our aim was to analyze northward transport not over all the Japan Sea but across the subpolar frontal zone between 40N and 42N only.

4. Cited from the referee's report 4. Have the authors compared the intrusions of subtropical waters (figure 6) with Sea Surface Temperature or Ocean color satellite images? It could help you to prove with observations such dynamical features? Our

response No.

Abstract: 5. Cited from the referee's report 1. The authors repeat several times "Lagrangian indicators (line 2)", Lagrangian maps (line 7) without specifying the Lagrangian diagnosis used in the study. Our response The text, describing Lagrangian indicators and maps, has been added to Sec.2 as follows: "Each water parcel can be attributed to temperature, salinity, density and other properties which characterize this volume as it moves. In addition, each water parcel can be attributed to more specific characteristics which are trajectory's functions called "Lagrangian indicators". They are, for example, a distance passed by a fluid particle, its displacement from an original position, time of residence of fluid particles inside a given area and others. The Lagrangian indicators contain information about the origin, history and fate of the corresponding water masses. Lagrangian maps are plots of Lagrangian indicators versus particle's initial positions. A studied area is seeded with a large number of tracers whose trajectories are computed for a given period of time to get the field of a specific Lagrangian indicator in this area. Finally, its values are coded by color and represented as a map in geographic coordinates."

6. Cited from the referee's report 3. Sentence in lines 5-6. It is hard to understand how the eddies open the gates due to their suitable dispositions. What does "suitable dispositions" means? Our response We mean by a suitable disposition of mesoscale eddies such a configuration where one of the eddies gains an amount of subtropical water from the south, wraps it around and transports that water to the north where another eddy is appropriately situated to do the same. This situation is documented in the paper with the help of different kinds of Lagrangian maps in Figs.6 and 7 and validated by tracks of available drifters. For example, a vortex street with four anticyclones has been formed in the fall of 2005 to the north of the Subpolar Front in the western part of the sea. Their centers are marked in Fig. 6a by the elliptic points (triangles) with the coordinates 39.1N, 131.5 E; 39.3N, 130.1E; 40.8N, 131.4E and 41.7N,130.8E. It is explained on p.11 (L.10-25), on p.11 (L.26-33) and on p.12 (L.11-18) in the text. The

gates open at those places where suitable dispositions of eddies and vortex streets appear time to time to facilitate the northward transport. They may appear due to different reasons, e.g., due to a seasonal variability of the current field, migration of frontal eddies etc.

7. Cited from the referee's report 2. Last two sentences: It's clear that something is happening because of the peculiarities of the advection velocity field, but which one? Please be more precise. Our response These peculiarities of the advective velocity field are described in detail in the last paragraph in Sec.3.1.

Introduction: 8. Cited from the referee's report 1. Figure 1 shows high values of northeastward velocity data in the TS (Tsugaru strait)? It means that, in average, the most of particles escape throughout this strait to the Pacific Ocean? Do the authors know if there is a realtionship between these features with the frontal-pair-vortex systems (AC-C) located at the entrance of the strait? Our response The high velocity in the strait does not mean that all the particles escape throughout this strait to the Pacific Ocean because it is compensated by a narrowness of the strait. We did not estimate the number of particles escaping to the Pacific Ocean and a connection of the AC-C vortex pair to that escape. 9. Cited from the referee's report 2. Please move to the first paragraph (where you are describing the bathymetry) the sentence in line 19 (page 1) with the reference to the figure showing the bathymetry. Our response Done.

10. Cited from the referee's report 3. Acronyms: The text is full of acronyms. - Could the authors explain what does SF mean (line 5, page 2)? Salinity fronts? - It could be great if the authors describe all the acronyms in the introductory part (second paragragh of introduction section) and not in the Results section (first paragraph of section 3.1) neither in the caption of Figure 1. Our response SF means a Subpolar front, but we removed that acronym from the revised text. We left in the main text only JS for the Japan Sea and AC and C for anticyclones and cyclones. The other acronyms in the text refer to the corresponding figures.

Data and methods: 11. Cited from the referee's report 1. The time step used for the Runge-Kutta integration is 1/1000. It means 0.001 days (1.5 minutes)? Why this such short time step? Have you compared the results using a longer time step, namely, 1 day (the time resolution of the Altimetry data?) Our response This time step is optimal to integrate ODEs. A 1 day step implies much more large numerical errors.

12. Cited from the referee's report 2. It is hard to understand the sentences from line 18 and line 21 page 4: " Trajectory of each : : :........in the northward transport only." Authors say that "the fixed the position and time of each tracers when they cross a given latitude between 37 and 43" and then they say that they "fix only the first crossing of a given latitude". Could the authors clarify this part of the methodology? Our response Some tracers could cross a few times the same zonal line in the northward direction. For example, if they belong to an eddy crossing that line. We fix only the first crossing of a given latitude by tracers.

13. Cited from the referee's report 3. line 21 page 4. Could the authors explain how a cell of AVISO can have two corners situated at land? As far as I am concerned a cell of AVISO only can be either land or ocean depending of the land mask. Maybe they mean two corners of the integration cell situated at land? Our response The AVISO grid contains points which are either at the land or in the sea. We mean by the cell a rectangle with 4 corners. There is no integration cells in integrating ODEs.

14. Cited from the referee's report 4. line 23-25. Could the authors read carefully and rewrite the paragraph? The sentence does not make sense. Our response It is rewritten to be "To simulate and analyze transport across the frontal area, we solve successively a few tasks which are numbered in the text in accordance with the following diagrams and Lagrangian maps."

15. Cited from the referee's report 5. Could the authors provide a description of the methodology used to compute and classify the fixed points of the velocity field in elliptic and hyperbolic points? The velocity field in the centers of the eddies is different to zero.

Could the streamlines of saddle nodes be used to localize fronts? Our response The stagnation points are zeroes of the interpolated velocity field at a fixed moment of time. They could be distinguished by eigenvalues of the evolution matrix. A point with zero velocity always exists at the eddy's center in a steady velocity field. The streamlines of saddle nodes could be really used to localize fronts but approximately only, because it's not obligatory for a real front to pass through an instantaneous hyperbolic point.

16. Cited from the referee's report 6. Please specify the number of drifters used in the study? Our response The number of drifters used in the study was 333. It has been added to the text.

Results and Discussion 17. Cited from the referee's report 1. The description given in the first paragraph is already provided in the introductory part (line 31 page 3 and lines 1, 2 page 4). Our response The first paragraph has been deleted and the second one was removed to Introduction.

18. Cited from the referee's report 2. First and second paragraph of this section are not results and they should be in the Introduction or in the Data and methods sections. Our response The first paragraph has been deleted and the second one was removed to Introduction.

19. Cited from the referee's report 3. Second paragraph page 7. Eulerian (time average of northward velocities, Fig. 2b) with Lagrangian (particles trajectories crossing different latitudes, Fig. 2a) diagnosis are compared. Authors state that both diagnosis are equivalent because the transport is determined by local advection. It means that the dynamics is controlled by the local scales ("small scales") and not by the large scales? Our response In the Lagrangian approach the dynamics is always local because it is governed by advection equations (1). The local advective velocity field determines where fluid particles cross a given latitude. However, the relative number of such crossings in different meridional peaks in Fig.2 is defined by a large-scale dynamics and initial conditions.

20. Cited from the referee's report 4. Do Fig. 3 show zonal cross-sections of Fig. 2 for four latitudes? If so, why the authors do not relate both figures in the manuscript, it could help to the reader. I do not know what does means "central part"? Authors state that "the correlation is rather good for western and eastern parts but not for the central one" (line 31-32, page 7), however, I see a good correlation between virtual and real drifters in the central part of the Japan Sea (see minima in Fig. 3 at 134E-136E). Moreover, this correlation is stronger in the north part (higher latitudes) than in the southern. Maybe because of the number of available drifters? Could the authors specify the number of drifters? Have the authors values of the correlations coefficients? Our response Yes, Fig.3 is a zonal cross sections of Fig. 2 for four latitudes. The first two paragraphs in Sec.3.2 have been rewritten to be: "Now let's look more carefully at the meridional distribution of subtropical tracers crossed the Subpolar Front for the whole period of simulation. We choose for reference four zonal lines along the AVISO grid at \N{42.125}, \N{41.875}, \N{40.125} and \N{39.875}. They are shown in Fig.~\ref{fig3} by solid curves with superimposed meridional distributions of the averaged northward AVISO velocity (arrows). The number of crossings of those latitudes by available 333 drifters is shown by dashed curves. The correspondence between the peaks in the meridional distributions of the tracers, drifters and the averaged northward AVISO velocity is rather good for all the chosen zonal lines confirming their direct connection. However, the comparison with drifters should be taken with care because of a comparatively small number of available drifters. Drifters, are not ideal passive tracers, and their motion is subjected to submesoscale features which were not caught by altimetry-derived data. Moreover, the drifters have not been launched at the zonal line \N{37} like artificial tracers in simulation. Their launch sites for more than 20 years have been distributed rather randomly over the basin."

21. Cited from the referee's report 5. With regard to the bad correlation with drifters (line 32-33 page 7), other explanation could be because drifters movement is due also to submesoscale features which are not provided by velocities derived from altimetry. Our response We corrected the corresponding sentences as indicated above.

22. Cited from the referee's report 6. Line 2 page 8. How does the authors know that drifters have been launched randomly over the basin? Could they provide references on these drifters experiments explaining these random releases? Our response 333 drifters have been launched for 20 years at different places. For our purposes, their releases could be considered to be random. As to references on these drifters experiments, it would take too much place in the Reference list and has no direct relation to our study.

23. Cited from the referee's report 6. Drifters are not launched at 37 N but at least they have to cross this latitude to take them into account in the meridional transport computations. Our response It is hardly possible to compare drifter's tracks with tracks of individual synthetic tracers in the AVISO velocity field by the reasons mentioned in the respond No.20. We use available drifters in order to demonstrate that they, like synthetic tracers, prefer to cross given latitudes at specific places (gates).

24. Cited from the referee's report 7. Second paragraph is confused. I do not know how to differentiate gates and barriers from Fig 3. Minima and maxima of numbers of tracers?. Could the authors clarify this point? Our response Yes, local maxima and minima of the number of tracers correspond to gates and barriers, respectively. It is clarified in the revised text as follows "The local maxima and minima of the distribution function correspond to gates and conditional barriers, respectively."

25. Cited from the referee's report 8. It is difficult to deduce from Figure 4 the parameter chosen by the authors to define the size of gates and barriers. Our response The size of the gates in Fig.4 is defined by the minima of the distribution function $N(\lambda f)$ at the latitude 42N.

26. Cited from the referee's report 9. Line 1-2, page 11. It's hard to understand what the authors mean in this sentence. Our response That's our statement mentioned by the referee: "Thus, the northward transport of subtropical water across the Subpolar Front occurs by a portion-like manner. Specific oceanographic conditions may arise in

a given area and at a given time which produce a large-scale intrusion of subtropical water to the north by means of mesoscale eddies to be present there." It is described just above that sentence in the text why it occurs by a portion-like manner. By specific oceanographic conditions providing a large-scale intrusion of subtropical water to the north, we mean mainly suitable dispositions of frontal mesoscale eddies. This point was clarified in our respond No.6 and in the corresponding place in the text.

27. Cited from the referee's report 10. Line 3, page 11- This motivation has been included in the introductory sections. Please avoid repetitions. Our response OK, we deleted that.

28. Cited from the referee's report 11. Line 6, page 11. Definition of Lagrangian Maps is not clear: I really do suggest that the authors rewrite it. Our response The following text, describing Lagrangian indicators and maps, has been added to Sec.2: "Each water parcel can be attributed to temperature, salinity, density and other properties which characterize this volume as it moves. In addition, each water parcel can be attributed to more specific characteristics which are trajectory's functions called "Lagrangian indicators". They are, for example, a distance passed by a fluid particle, its displacement from an original position, its travel time and others. The Lagrangian indicators contain information about the origin, history and fate of the corresponding water masses. Lagrangian maps are plots of Lagrangian indicators versus particle's initial positions. A studied area is seeded with a large number of tracers whose trajectories are computed for a given period of time to get the field of a specific Lagrangian indicator whose values are coded by color and represented as a map in geographic coordinates."

29. Cited from the referee's report 12. Fig 6 correspond to maps of residence times computed backward in time. They are the incoming times and it has been already used for several authors: - Lipphardt, B., Jr., Small, D., Kirwan, A., Jr., Wiggins, S., Ide, K., Grosch, C., and Paduan, J.: Synoptic Lagrangian maps: Aplication to surface transport in Monterey Bay, J. Mar. Res., 64, 221– 247, 2006. - H. Gildor, E. Fredj, J. Steinbuck, S. Monismith. Evidence for submesoscale barriers to horizontal mixing in the ocean from

current measurements and aerial photographs. Journal of Physical Oceanography, 39, 1975–1983, 2009. - HernĐśndez-Carrasco, I., LŇČpez,C., Orfila, A., and HernĐśndez-GarcĐ¡a, E.; Lagrangian transport in a microtidal coastal area: the Bay of Palma, island of Mallorca, Spain. Nonlin. Processes Geophys., 20, 921–933, 2013. Our response Formally, Fig 6 is not a map of residence times. It is a map of travelling time T that took for subtropical tracers to reach to a specific date their locations on the map from the latitude 37N. We have added the first and third references to the text. The second one has no relation to the residence maps.

30. Cited from the referee's report 13. It could be great if authors include a reference when they say that Peripheries of the mesoscale eddies in the ocean are known to be transport pathways for larvae, fish : : :..... (lines 5-6, page 12). Our response There is a vast literature on that subject. We have referred to some in the revised text (see, e.g., Cotte2010,P13,Prants2014c and references therein).

31. Cited from the referee's report 14. Sentence in line 7: ": : :..kind of transport for heat-loving organisms to reach the southern coast of Russia". It means that the coast of Russia is warm? Our response It means that some subtropical (=heat-loving) organisms are able to reach the southern coast of Russia. Most of people are rather heat-loving organisms, but they could reach the Earth's poles using some transport means.

32. Cited from the referee's report 15. The paragraph (lines 8-15) could be improved if the authors reorganize the text. Lines 11-13 (" The red and green : : :....respectively") could be move in before line 10 ("In the beginning of September : : :...pulled to the north.") Our response Done. 33. Cited from the referee's report 16. Line 3, page 13. It seems that this is the mechanism to explain the intrusions of subtropical waters (Figure 6). The manuscript could be improved if authors are more explicits. Our response We clarified in our respond No.6 how frontal eddies could facilitate the northward transport of subtropical water. It is also described in Sec.3.2 for the western, central and eastern gates and in the Supplementary material.

34. Cited from the referee's report 17. Is the displacements of the tracers (D) a Lagrangian diagnosis developed by other authors, by means of the function M, in order to obtain the Lagrangian Coherent Structures?: - Mendoza, C. and Mancho, A. M.: The hidden geometry of ocean flows, Phys. Rev. Lett., 105, 038501, doi:10.1103/PhysRevLett.105.038501, 2010. Our response The definition of particle's displacement D has been added to Sec.2 as follows": The drift maps show by nuances of the grey color the finite-time displacement of tracers, $D$, that is a distance between final, $(\lambda_f,\,\varphi_f)$, and initial, $(\lambda_0,\,\varphi_0)$, positions of advected particles on the Earth sphere with the radius $R\_E$ \begin{equation} D\equiv R_E\arccos[\sin \varphi_0 \sin \varphi_f +\cos \varphi_0 \cos \varphi_f \cos (\lambda_f - \lambda_0)]. \label{drift} \end{equation} It is not the length of a trajectory. 35. Cited from the referee's report 18. Line 7, page 13. " so, the black tracers : : :......white ones" I do not understand what the authors mean with "displaced". What is the difference between white and black colors? The length of the trajectory? Our response See please the preceding respond. The black particles have displaced for an integration time much farther that the white ones.

36. Cited from the referee's report 19. Line 14, page 13: "....pro-pulsed to the northwest". Do the authors mean "northeast" instead of "northwest"? It seems that particles are not transported northwestward by the frontal-pair-vortex system but northeastward. Our response Corrected. Thank you.

37. Cited from the referee's report 20. I do not understand the last paragraph from line 14 to line 18, page 13. Are the authors describing the drifter trajectories? Our response Yes, we describe the drifter's track shown by green circles in Fig.7.

38. Cited from the referee's report 21. Line 24: Simulations with "imperfect" AVISO is compared with satellite, drifter and in situ observations. What kind of satellite product has been used to be compared with AVISO? When comparing with drifter trajectories you should take into account that drifter velocities has to be different than the geostrophic velocities derived from AVISO by definition. Our response We removed all

that paragraph.

39. Cited from the referee's report 22. Line 25-30. This sentence has been mentioned three times in the manuscript. Please avoid repetitions. Our response We removed all that paragraph.

40. Cited from the referee's report 23. Line 31: "Nobody knows, of course true velocity field in the real ocean" This sentence is not necessary, please remove it. Our response Removed.

41. Cited from the referee's report 24. Line 32: "The AVISO velocity field has errors as compared with an unknown "true" velocity field...". This sentence is not logical: in my opinion anything can not be compared with things that are unknown, because they are unknown. One could say that the unknown part of the velocity field could be simulated by adding noise in the velocity data. Our response Corrected to be: "The AVISO velocity field has errors as compared with a "true" velocity field. The difference could be simulated by adding a noise $\Delta (u,v)$ in the velocity data."

42. Cited from the referee's report 25. Line 34, page 13. What does "To which extent one can trust to them" mean? Our response This sentence was removed.

43. Cited from the referee's report 26. Line 19, page 4. This study of adding errors in the velocity field to compute the trajectories has been performed for the FTLE and also for FSLE. Our response Corrected to be "...by the finite-time and finite-size Lyapunov techniques."

Conclusions 44. Cited from the referee's report 1. I have found many repetitions that have been copied from the body of the manuscript. I suggest that the authors rewrite the whole conclusions section. Our response We removed all the not necessary repetitions in the revised version.

45. Cited from the referee's report 2. Please remove references to the Figures. Our response It seems to us that it would be more convenient to the reader to get a reference

to the figure which illustrates the corresponding result.

Technical comments: 46. Cited from the referee's report 1. Please remove "there" (last word of abstract, line 11, page 1) Our response ĐđĐž

47. Cited from the referee's report 2. Why "Sea" (line 5 page 2) is in capital letter? Our response Corrected.

48. Cited from the referee's report 3. Line 6 page: 3 replace "to the coast" with "in the coast" Our response Corrected.

49. Cited from the referee's report 4. Line 10, page 3: Remove "across it". Our response Corrected.

50. Cited from the referee's report 5. Line 11 page 3: in " we use the altimetry data" remove "the". Our response Corrected.

51. Cited from the referee's report 6. Line 14 page 3. Remove "based on altimetry data". It has already been stated several times in the text. Please avoid repetitions. Our response Done.

52. Cited from the referee's report 7. Line 25, 26, page 7, It is not necessary to specify the longitudes for the zonal cross-section but only with the latitude the reader can localize the zonal lines. Our response Corrected.

53. Cited from the referee's report 8. Please read carefully the lines 28 in page 7 and rewrite the text: " The number of crossing : : :....dashed curves" by for instance "The number of available drifters crossing the given latitudes are shown : : :......". Our response Corrected to be: The number of crossings of those latitudes by available 333 drifters is shown by dashed curves.

54. Cited from the referee's report 9. There are many "to the" through the manuscript that should be replaced with "at the" or "in the" Our response Done.

55. Cited from the referee's report 10. Line 1 , page 9. What does "task 2" mean?

Authors use "task X" throughout the manuscript to refer to the computations given in the Data and methods section. It could be great if it is stated in the methodology section that the computation X correspond to task X. (Although task is usually used in the context of a project and not in a paper). Our response It is clarified in Sec.2 as follows: "To simulate and analyze transport across the frontal area, we solve successively a few tasks which are numbered in the text in accordance with the following diagrams and Lagrangian maps."

56. Cited from the referee's report 11. There are two reference for Prants et al, 2015. Our response There are no two references for Prants et al, 2015.

57. Cited from the referee's report 12. Line 1, page 12. Please remove the website of drifter data (it has been shown in Data and Methods) and also the number of the drifters. Our response Done. 58. Cited from the referee's report 13. Line 23, page 13. Remove "the" before AVISO and "we used" after AVISO. Our response Done. 59. Cited from the referee's report 14. Line 5, page 14. Please replace the sentence " So , locations of the preferred : : :...are possible" with "So, locations of the preferred transport pathways are not expected to be changed significantly". Our response Corrected. 60. Cited from the referee's report 15. Line 9, page 14. Please replace "Say" with "For example". Our response Done.

61. Cited from the referee's report 16. Line 14, page 14. Replace "a plenty of " by "the presence of numerous". Our response Done.

62. Cited from the referee's report 17. Line 16, page 14. Replace "As to" by "The" and remove "it" Our response Done.

63. Cited from the referee's report 18. Line 17, page 14. Add "by analyzing" just before "how and additional". Our response Done.

64. Cited from the referee's report 19. Line 18, page 14. remove "modelling unknown corrections to the AVISO velocity field". It has already been mentioned. Our response

Done.

65. Cited from the referee's report 20. Line 19, page 14. Add "Finite Size Lyapunov Exponents" after "Finite Time Lyapunov Exponents" Our response Done. See please our respond No.43.

Please also note the supplement to this comment:
http://www.nonlin-processes-geophys-discuss.net/npg-2016-67/npg-2016-67-AC3-supplement.pdf